# Divergent Seasonal Patterns of Qinghai Spruce Growth with Elevation in Northwestern China

Yanfang Wan [1], Pengtao Yu [1,*], Xiaoqing Li [2], Yanhui Wang [1], Bin Wang [1,3], Yipeng Yu [1], Lei Zhang [4,5], Xiande Liu [6] and Shunli Wang [6]

[1] Ecology and Nature Conservation Institute, Chinese Academy of Forestry, Key Laboratory of Forest Ecology and Environment of National Forestry and Grassland Administration, Beijing 100091, China; wanyf1993@163.com (Y.W.); wangyh@caf.ac.cn (Y.W.); wangbinlky@163.com (B.W.); god891006@caf.ac.cn (Y.Y.)

[2] School of Politics and Public Managements, Qinghai Nationalities University, Xining 810000, China; lixiaoqing826@163.com

[3] College of Resources and Environmental Engineering, Ludong University, Yantai 264025, China

[4] State Key Laboratory of Urban and Regional Ecology, Research Center for Eco-Environmental Sciences, Chinese Academy of Sciences, Beijing 100085, China; leizhang_st@rcees.ac.cn

[5] University of Chinese Academy of Sciences, Beijing 100049, China

[6] Academy of Water Resource Conservation Forests of Qilian Mountains in Gansu Province, Zhangye 734000, China; liuxiande666@163.com (X.L.); wangshun123_78@163.com (S.W.)

*  Correspondence: yupt@caf.ac.cn; Tel.: +86-10-62889562

**Abstract:** Dryland montane forests are important agents for soil and water resource conservation. The growth of these forests under climate warming is strongly affected by local environmental factors. However, how environmental factors impact intra-annual stem growth dynamics across environmental gradients in these regions remains unclear. This work focused on assessing seasonal patterns of stem growth across different elevations and how environmental factors impact stem growth in the Qilian Mountains, northwestern China. The stem growth of 50 Qinghai spruce trees was monitored for two years across an elevation gradient from 2500 m to 3300 m above sea level (a.s.l.). We found that growth initiation occurred later as the elevation increased, and growth commenced when elevation-specific temperature thresholds were reached. However, growth cessation presented large elevational differences: cessation occurred much earlier at low elevations (2500 m and 2700 m a.s.l.). Exceptionally early growth cessation occurred predominantly at 2700 m a.s.l., which was correlated with seasonal drought/insufficient rainfall and low soil moisture occurring since mid-July 2015. Temperature and soil moisture were the key factors governing the daily rate of stem growth in the beginning, rapid growth, and end stages. Overall, due to effects of seasonal drought and low temperature on growth cessation and growth rate, the annual growth of Qinghai spruce was rather low at both low (2500–2700 m a.s.l.) and high (3100–3300 m a.s.l.) elevations; middle elevations (approximately 2900 m a.s.l.) might be the most favorable Qinghai spruce growth. Our results implied that tree growth will likely decline at low elevations and that the optimal elevation for Qinghai spruce growth in northwestern China is expected to shift upward under future climate warming.

**Keywords:** *Picea crassifolia*; tree growth; seasonal pattern; environmental factors; elevation gradient

## 1. Introduction

Climate changes, particularly dramatic warming and severe drought events, have occurred around the world [1], most notably in northern China [2,3]. These changes exert a strong influence on forest dynamics and growth [4–6]. Therefore, to assess the impacts of future climate change on forest growth, it is crucial to study the climatic forcing of tree growth at annual and seasonal time scales. However, a number of studies about growth-climate relationships were mostly based on tree-ring-based investigations [7–10]. Such studies do not fully explain the effects of climate conditions on tree growth and do

not capture seasonal growth changes and relevant growth-limiting factors. Hence, studies on seasonal growth responses to climate changes are urgently needed, it can help our understanding of tree growth at annual or multiyear time scales.

Elevation, as a natural simulant of future climates [11], has a strong impact on local climate conditions, and seasonal patterns of stem growth and climate–growth relationships may vary with elevation, such as variations in temperature and precipitation [12–14]. Many studies have reported that intra-annual stem growth for the same tree species had divergent across elevations; i.e., the growth initiation was linearly delayed and nearly synchronous with increasing elevations, which was primarily controlled by temperature [14–17]; however, the growth cessation was no consensus across elevations, which was predominately controlled by photoperiod and an earlier cell differentiation [15,18,19], while was influenced by soil moisture deficit in arid and semi-arid regions [17,20–22]. Although there is still much uncertainty about intra-annual stem growth at different elevations, how seasonal patterns of stem growth is controlled by climatic and soil conditions at different elevations has rarely been studied in dryland montane forests of northwestern China.

The Qilian Mountains in northwestern China constitute a key nourishing and cherished water source area [23]. Qinghai spruce (*Picea crassifolia* Kom.) tree growing in this region is a native evergreen conifer growing at an elevation of 2500–3300 m a.s.l. [7,9]. Earlier studies showed that daily and seasonal variation in stem growth of Qinghai spruce trees at the single scale and among tree classes, and that stem growth initiation was governed by air temperature and/or soil temperature [20,23–25], and that growth cessation was affected by soil moisture under drought conditions [20]. However, the intra-annual variation in stem growth across elevations remains poor and the environmental factors impacts are not completely understood.

In this study, we hypothesized that there was a uniform intra-annual variation grow pattern at all elevation, i.e., growth initiation was controlled by temperature, while growth cessation was affected by soil moisture at all elevations. To test the above hypothesis, we selected five Qinghai spruce forest plots across elevations in the Qilian Mountains, and we monitored stem diameter and environmental conditions in the growing seasons of 2015 and 2016. The objectives of this study were to: (a) assess the divergent seasonal patterns of stem growth across different elevations, (b) determine the thresholds of environmental factors that determine the seasonal growth patterns, and (c) determine how environmental factors affect the daily stem growth rate at different growth stages. Our results will contribute to improving the understanding of the intra-annual stem growth across elevations and to predicting future forest development in these regions under changing environmental conditions.

## 2. Materials and Methods

### 2.1. Study Area

The study was conducted in the Pailugou watershed of the Qilian Mountains, Gansu Province (38°22′–38°35′ N, 100°17′–100°19′ E). The Pailugou watershed covers an area of 2.85 km$^2$; the area has an elevation range of 2500–3800 m a.s.l. and an arid and semiarid continental climate, with a mean annual air temperature of 1.6 °C, a mean annual precipitation of 435.5 mm, a basin evaporation rate of 1081.7 mm·year$^{-1}$ and a mean annual relative humidity of 60% (the mean during 1994–2014 at the Xishui weather station at 2600 m a.s.l.) [20].

In the Pailugou watershed, montane forests are mainly distributed on shaded or semi-shaded slopes at an elevation of 2500–3300 m a.s.l., and the forest is dominated by Qinghai spruce. The stand density decreases from 2800–2000 trees·ha$^{-1}$ to only approximately 300 trees·ha$^{-1}$ with increasing elevation. The soil is thicker and the sand content lower at elevations from 2500 to 2900 m a.s.l., but the soil layer is relatively thin and unevenly distributed at elevations from 3000 to 3300 m a.s.l. [9]. The main shrub community include *Salix Gilashanica* C. Wang et P. Y. Fu, *Caragana jubata* (Pall.) Poir., and *Potentilla fruticosa* L. [26]. The dominant understory herbs are *Stipa capillata* L.,

*Polygonum viviparum* L., and *Pedicularis* spp. Moreover, moss is widely distributed under Qinghai spruce forests, and the main moss species is *Abietinella abietina*.

### 2.2. Experimental Design

In 2015, five plots (20 m × 20 m in size) were selected across an elevation gradient at specific elevations of 2500 m, 2700 m, 2900 m, 3100 m and 3300 m (Figure 1). The plots at elevations of 2500 m and 3300 m were located in the lower- and upper-forest line ecotones, respectively. All five plots were located on semi-shaded hillsides with similar slope gradients of approximately 27–41°. The soil type of the five plots was a mountainous gray cinnamon soil; this soil texture was considered a loam, with a sand content of 34–58% [9], and the soil thickness was 50–80 cm.

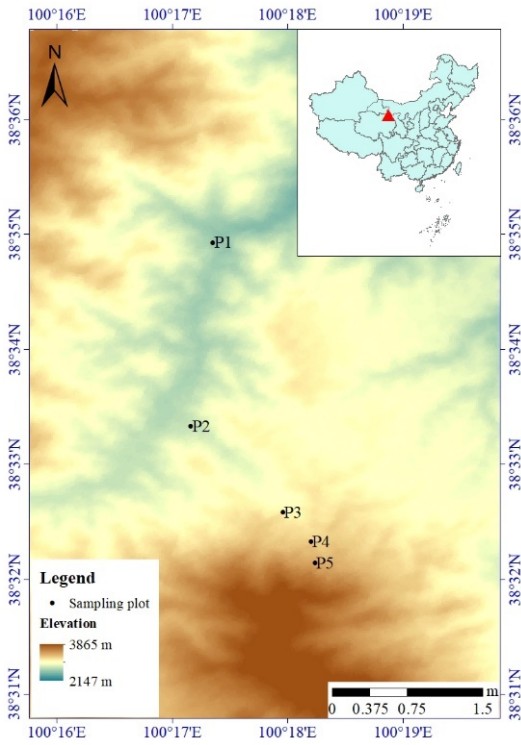

**Figure 1.** The digital elevation model (DEM) of the study area showing the locations of Qinghai spruce plots. P1 to P5 represent the locations of five plots at elevations of 2500 m, 2700 m, 2900 m, 3100 m and 3300 m, respectively.

The forest in each plot consisted of Qinghai spruce trees. For each tree in the plots, the diameter at breast height (DBH), tree height, and canopy width were recorded. Basic information concerning elevation, soil depth, aspect, and the slope of each plot was also recorded during field work (Table 1).

**Table 1.** Basic characteristics of the five Qinghai spruce plots (mean value ± standard deviation).

| Plots | Elevation (m a.s.l.) | Soil Thickness (cm) | Aspect (°) | Slope (°) | Density (Trees·ha$^{-1}$) | Average DBH (cm) | Average Tree Height (m) | Average Canopy Diameter (m) |
|---|---|---|---|---|---|---|---|---|
| P1 | 2500 | 80 | NE12 | 41 | 1375 | 11.8 ± 8.3 | 6.9 ± 5.0 | 3.3 ± 1.6 |
| P2 | 2700 | 70 | NE30 | 27 | 2100 | 11.8 ± 6.5 | 8.4 ± 3.8 | 3.0 ± 1.0 |
| P3 | 2900 | 60 | NE24 | 32 | 2000 | 13.6 ± 7.7 | 8.2 ± 3.7 | 3.2 ± 1.3 |
| P4 | 3100 | 50 | NE25 | 32 | 825 | 13.5 ± 8.3 | 6.9 ± 3.5 | 3.3 ± 1.4 |
| P5 | 3300 | 50 | NE32 | 35 | 375 | 14.2 ± 8.9 | 5.6 ± 2.9 | 4.2 ± 1.9 |

*2.3. Data Collection*

2.3.1. Dendrometer Record Collection

In whole forest, Qinghai spruce trees were divided into three size classes according to their DBH, i.e., large trees (DBH > 22.5 cm), medium trees (12.5 cm < DBH ≤ 22.5 cm) and small trees (4.0 cm < DBH ≤ 12.5 cm) [27]. A total of ten sample trees in each plot were selected, including four large trees, three medium trees and three small trees. The characteristics of sample trees of each size class are shown in Table 2. There was not significant difference of the DBH of sample trees among the five plots (Kruskal–Wallis test, $p > 0.05$).

**Table 2.** Characteristics of sample trees of each size class in the five Qinghai spruce plots (mean value ± standard deviation).

| Plots | Small Trees | | | Medium Trees | | | Large Trees | | |
|---|---|---|---|---|---|---|---|---|---|
| | Average DBH (cm) | Average Height (m) | Average Canopy Diameter (m) | Average DBH (cm) | Average Height (m) | Average Canopy Diameter (m) | Average DBH (cm) | Average Height (m) | Average Canopy Diameter (m) |
| P1 | 7.1 ± 1.8 | 3.7 ± 1.6 | 2.3 ± 0.2 | 21.4 ± 0.3 | 10.8 ± 1.1 | 5.2 ± 0.2 | 28.3 ± 3.9 | 13.0 ± 1.3 | 5.6 ± 0.1 |
| P2 | 8.5 ± 3.4 | 6.9 ± 3.2 | 2.7 ± 0.8 | 16.0 ± 0.8 | 11.8 ± 1.2 | 3.6 ± 0.4 | 30.6 ± 1.6 | 14.6 ± 1.5 | 5.7 ± 0.7 |
| P3 | 9.2 ± 3.0 | 6.3 ± 2.0 | 2.8 ± 0.5 | 15.3 ± 4.0 | 9.7 ± 3.7 | 3.7 ± 0.4 | 26.8 ± 5.0 | 14.0 ± 0.5 | 5.0 ± 0.7 |
| P4 | 8.1 ± 3.0 | 4.8 ± 1.9 | 2.2 ± 0.8 | 15.3 ± 2.7 | 8.6 ± 0.8 | 3.6 ± 0.6 | 30.8 ± 4.0 | 12.9 ± 0.9 | 6.0 ± 1.2 |
| P5 | 7.3 ± 2.2 | 3.1 ± 0.4 | 2.6 ± 0.7 | 16.9 ± 2.6 | 6.8 ± 0.3 | 4.6 ± 0.6 | 26.2 ± 2.5 | 9.3 ± 1.6 | 6.2 ± 0.6 |

The stem diameter growth of 50 sample trees was measured at 1.3 m above the ground using band dendrometers. The dendrometers (Ecomatik, Munich, Germany), with a resolution of 0.1 mm, consists of a spring and a flexible stainless-steel band with a scale. To minimize disturbance caused by the bark expansion and contraction, the dead bark was lightly removed, and then the dendrometer was installed on the trunk [20]. The scale of the dendrometers varied with trunk changes. The initial stem diameter of individual sample tree was observed on 1 May. Data were collected at approximately 5–10 days intervals to determine whether the stem diameter had changed. The whole experiment was conducted across different elevations from 1 May to 31 October in both 2015 and 2016. Data at elevations of 3100 m and 3300 m, at which the equipment failed in 2016, were eliminated.

The cumulative stem diameter growth ($G_0$, μm) was calculated by Equation (1):

$$G_0 = D_i - D_0 \tag{1}$$

where $D_i$ is the daily stem diameter on day $i$ and $D_0$ is the initial stem diameter on 1 May 2015 or 2016.

To minimize the impacts of varying tree size, this study used the mean $G$ of the measured cumulative stem diameter growth ($G_0$) of ten sample trees covering the DBH range in each plot.

2.3.2. Weather and Soil Data Collection

Meteorological data were collected using an automatic weather station (CR3000, Campbell Scientific Inc., Logan, UT, USA) with sensors height of approximately 2 m, installed in an open area at 2700 m a.s.l. Meteorological data included air temperature (HMP115A, $T_a$ (°C)) and precipitation (TE525MM, $P$ (mm)) during the growth periods of 2015 and 2016 (May–October). The elevation gradients of daily temperature and precipitation were calculated using the data from this base weather station. With increasing elevation, the daily $P$ showed an increasing rate of approximately 4.95%·100 m$^{-1}$ of elevation, while the daily $T_a$ decreased at a rate of approximately 0.58 °C·100 m$^{-1}$ of elevation in the Pailugou watershed [28–30], as shown via Equations (2) and (3) below.

$$P_a = P_{2700} \times (1 + 4.95\%)^{\frac{H_a - 2700}{100}} \tag{2}$$

$$T_a = T_{2700} - 0.58 \times \frac{H_a - 2700}{100} \tag{3}$$

where $P_a$ and $T_a$ are the daily precipitation and daily air temperature at any elevation $H_a$ (m a.s.l.).

In each plot, the volumetric soil moisture content ($M_s$, %) and soil temperature ($T_s$, °C) of the 0 to 10 cm, 10 to 20 cm, 20 to 40 cm, and 40 to 60 cm soil layers were continuously monitored using soil moisture and temperature sensors (5-TE, Decagon, Pullman, WA, USA). Data were stored at 10 min intervals in a data logger (EM50, Decagon, Pullman, WA, USA). Since the active roots of Qinghai spruce trees were focused on the 0 to 60 cm soil layer, the weighted averages of $M_s$ and $T_s$ of the 0 to 60 cm soil layer were calculated using the measured $M_s$ and $T_s$ of each soil layers.

*2.4. Data Analysis*

2.4.1. Seasonal Growth Pattern Assessment

The Gompertz function was used to determine seasonal patterns of stem growth. This function is one of the most commonly used models [18] because of its flexibility and the asymmetrical shape of the resulting curve [31]. In this study, to characterize the complete seasonal growth pattern (May–October), the cumulative stem diameter growth curves of Qinghai spruce trees were fitted using the Gompertz function [18] (Equation (4)):

$$Y = A \, exp\left(-e^{(\beta - \mathrm{k}t)}\right) \tag{4}$$

where $Y$ is the cumulative stem diameter change, $A$ is the upper asymptote, $\beta$ is the *x*-axis placement parameter, $k$ is the rate of change parameter, and $t$ is the day of year (DOY).

The parameters of the Gompertz function were estimated by the ordinary least squares method with the MODEL procedure (1stOpt software, 7D-Soft High Technology, Inc., Beijing, China). The use of this model has the advantage of smoothing the measured stem diameter growth records. After smoothing, the daily rate of stem growth ($G_r$, μm·day$^{-1}$) curves were obtained by the first-order derivation of the simulated cumulative stem changes.

The mean DBHs and tree heights of ten sampled trees were similar to those of the plots. Therefore, the averages of ten sample trees at each elevation were analyzed. To assess interannual variability, we modeled the averaged stem diameter changes of ten sample trees at each elevation for each year ($n = 2$). The timing of growth initiation and cessation was determined as DOY when the daily growth rates exceeded the threshold value of 2 μm·day$^{-1}$, which corresponded to the change in cumulative stem diameter growth [20].

2.4.2. Analysis of the Relationship between Stem Growth and Environmental Factors

Environmental factors have a strong impact on the seasonal patterns of stem growth. Temperature and soil moisture are generally the key influencing factors for initiation and cessation of stem growth [20]. Therefore, we selected the $T_a$, mean $T_s$ and mean $M_s$ of the 0 to 60 cm soil layers to determine the thresholds of environmental factors for the initiation and cessation of stem growth of Qinghai spruce trees. According to the growth initiation and cessation data, we selected the $T_a$, $T_s$ and $M_s$ data before and after 4 days (9 days in total) and then performed one-way analysis of variance (ANOVA). If the results of the ANOVA are significant, the environmental factor in question is not at the environmental threshold that governs stem diameter growth; however, if the results of the ANOVA are not significant, the environmental factor is likely to control the threshold of only the stem diameter growth [32].

To better analyze the relationship between environmental factors and stem diameter variation over the growing season (May–October), the dendrometer data for each elevation were divided into three stages according to growth changes: the beginning stage (stage 1), the rapid growth stage (stage 2) and the end stage (stage 3) [20]. The corresponding environmental factors were also divided into three stages. Spearman correlation coefficients between the daily rate of stem growth ($G_r$) (obtained by using the Gompertz functions) and

environmental variables ($T_a$, $T_s$, $P$ and $M_s$) were calculated using the Statistical Product and Service Solutions (SPSS), version 19.0 (IBM Inc., Chicago, IL, USA).

## 3. Results

### 3.1. Environmental Variables and Gradients during the Growing Season across Different Elevations

In the Pailugou watershed, which is a mountainous watershed, the annual $P$ mostly occurs in the growing season; moreover, the temporal distribution of $P$ is uneven in the growing season (May–October). For example, at 2700 m a.s.l., the growing seasons $P$ were 76 and 88 rainy days in 2015 and 2016, with total $P$ of 369 mm and 333.7 mm, respectively, 84.2% (2015) and 92.0% (2016) of which had daily $P$ of less than 10 mm. The $T_a$ varied in the ranges of −5.5–22.4 °C and −7.6–23.3 °C, with averages of 10.1 ± 4.9 °C and 10.8 ± 5.8 °C in 2015 and 2016, respectively (Figure 2a,b). Normally, the $P$ increased and the $T_a$ decreased with increasing elevation. There were change rates of 0.58 °C·100 m$^{-1}$ of elevation for $T_a$ and 4.95%·100 m$^{-1}$ of elevation for the growing season $P$ when the elevation increased [28–30].

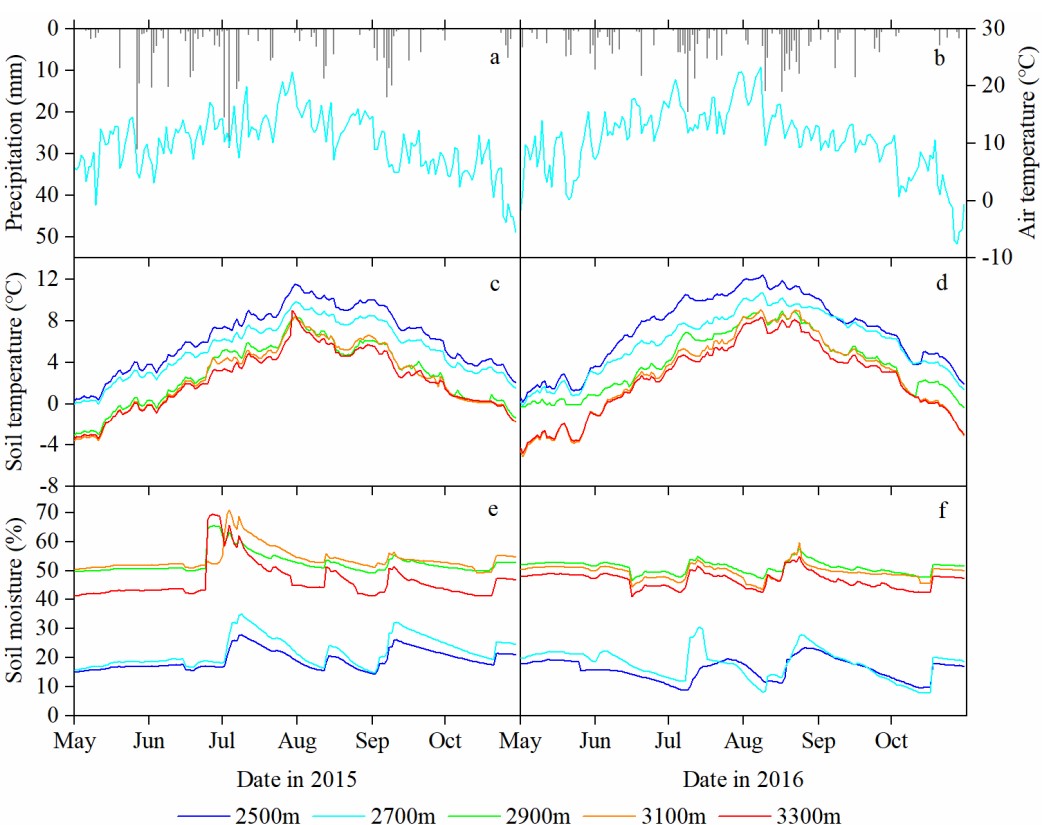

**Figure 2.** Daily variation of precipitation ($P$) and mean air temperature ($T_a$) at 2700 m a.s.l., and edaphic factors (mean soil temperature ($T_s$) and volumetric soil moisture content ($M_s$)) of the 0 to 60 cm soil layer at different elevations in the growing seasons of 2015 (**a,c,e**) and 2016 (**b,d,f**).

However, the elevation change pattern of $T_s$ differed significantly from that of $T_a$, and the pattern of $M_s$ was opposite that of $T_s$. For the $T_s$ and $M_s$, there is a critical line at the middle elevation (2900 m a.s.l.), i.e., there is markedly different change ratio both above and below this elevation. In 2015, the $T_s$ was approximately 6.0 °C at low elevations (such as 6.28 ± 2.98 °C and 5.23 ± 2.62 °C at 2500 m and 2700 m a.s.l., respectively), whereas it was approximately 2.5 °C at both middle and high elevations (2.86 ± 2.99 °C, 2.59 ± 3.12 °C and 2.27 ± 2.86 °C at 2900 m, 3100 m and 3300 m a.s.l., respectively) (Figure 2c). The $M_s$ was approximately 20% at low elevations (such as 19.27 ± 3.37% and 22.12 ± 5.03% at 2500 m and 2700 m a.s.l.), respectively, whereas it was approximately 50% at middle and high elevations (52.58 ± 3.57%, 53.97 ± 3.98% and 46.62 ± 6.36% at 2900 m, 3100 m and

3300 m a.s.l., respectively) (Figure 2e). In 2016, there were still $T_s$ and $M_s$ differences of 3 °C and 30% between elevation groups (below and above the elevations of 2900 m) and similar $T_s$ and $M_s$ levels across elevations in the same group (Figure 2d,f).

Temperature and soil moisture remained significantly different both two growing seasons. In May 2015, the $T_a$ and $T_s$ values (2700 m a.s.l., 8.5 °C and 1.6 °C, respectively) were slightly warmer than those in May 2016 (7.1 °C and 1.4 °C, respectively). From mid-July to August 2015, the $M_s$ continued to decline and reached a minimum value for the growing season, which can lead to a deficit in soil water hindering tree growth at 2500 m and 2700 m a.s.l. However, in July and August 2016, a different soil moisture pattern emerged in which the rainfall was uniformly distributed, leading to high levels of soil moisture at 2500 m and 2700 m a.s.l. Such environmental conditions likely promote differences in interannual tree growth.

### 3.2. Divergent Seasonal Patterns of Stem Diameter Growth with Changes in Elevation

Seasonal patterns of the measured and modeled cumulative stem growth and the daily growth rate for different elevations in both years are shown in Figure 3. The results of the Gompertz function, i.e., the modeled cumulative stem growth data, exhibited remarkable agreement with the measured growth data, with $R^2$ values greater than 0.98. Seasonal patterns of cumulative stem growth at all elevations exhibited an "S" shape in 2015 and 2016. Stem growth at all elevations started in May–June, peaked in July, and finally stopped in August–September.

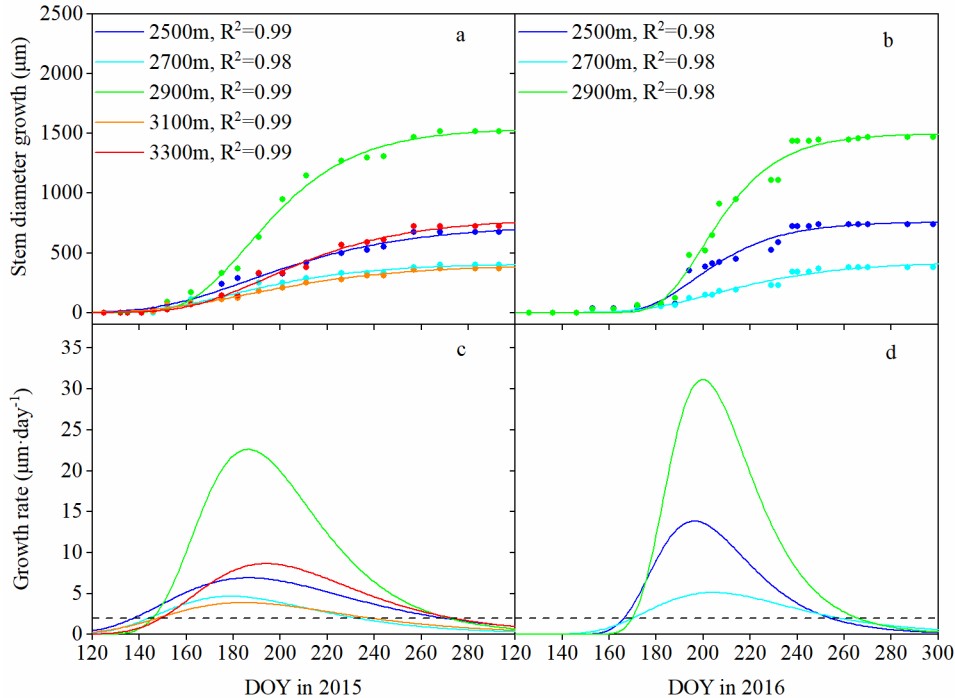

**Figure 3.** Gompertz function-modeled curves of the cumulative stem growth (**a**,**b**) and daily growth rate (**c**,**d**) at different elevations during the growing seasons of 2015 and 2016 (May–October). The dashed lines at the bottom panels indicate the daily growth rate (equal to 2 $\mu m \cdot day^{-1}$).

Seasonal patterns at different elevations showed divergent trends (Figure 3). The beginning of the growing season occurred later as the elevation increased; for example, in 2015, the occurrence of growth initiation ranged from May 18 (day of year (DOY) 138) at 2500 m a.s.l. to May 31 (DOY 150) at 3300 m a.s.l. (Table 3), yielding an onset delay of approximately 1.8 days·100 $m^{-1}$ of elevation ($R^2$ = 0.84, $p < 0.05$); in 2016, growth initiation at 2500 m a.s.l. occurred 5 days earlier than at other elevations (Table 3). However, there was a significant difference in growth cessation across elevations. Growth cessation occurred

much earlier at lower elevations; for example, in 2015, stem growth at 2700 m a.s.l. occurred 35 days earlier than at 2500 m, 2900 m and 3300 m a.s.l.; in 2016, stem growth at low elevations (2500 m and 2700 m a.s.l.) occurred 1 week earlier than it did at middle elevation (2900 m a.s.l.) (Table 3).

**Table 3.** Characteristics of seasonal growth patterns at different elevations in 2015 and 2016.

| Years | Elevation (m a.s.l.) | Timing of Growth Initiation (DOY) | Timing of Growth Cessation (DOY) | Growing Season Duration (days) | Maximum Growth Rate (μm·day$^{-1}$) | Day of Maximum Growth (DOY) | Cumulative Stem Growth (μm) |
|---|---|---|---|---|---|---|---|
| | 2500 | 138 | 269 | 132 | 6.907 | 186 | 675 |
| | 2700 | 145 | 233 | 89 | 4.658 | 179 | 400 |
| 2015 | 2900 | 146 | 271 | 126 | 22.616 | 186 | 1520 |
| | 3100 | 148 | 238 | 91 | 3.881 | 184 | 370 |
| | 3300 | 150 | 273 | 124 | 8.642 | 194 | 722 |
| | 2500 | 167 | 254 | 88 | 13.867 | 196 | 738 |
| 2016 | 2700 | 172 | 258 | 87 | 5.327 | 204 | 380 |
| | 2900 | 172 | 265 | 94 | 31.144 | 200 | 1470 |

Maximum stem growth at all elevations synchronously occurred in early July (DOY 179–194) in 2015 and in mid-July (DOY 196–204) in 2016 (Table 3). However, the maximum daily growth rate at 2900 m a.s.l. was higher than that at the other elevations, with values of 22.621 μm·day$^{-1}$ in 2015 and 31.504 μm·day$^{-1}$ in 2016, which were 2.2–5.8 times those at other altitudes. Accordingly, the cumulative stem diameter increases varied in accordance with a "unimodal" trend with increasing elevation, with a peak at 2900 m a.s.l. (1520 μm in 2015 and 1670 μm in 2016), which was 2.1–4.4 times that at other elevations.

### 3.3. Thresholds of Environmental Factors for the Seasonal Patterns of Stem Diameter Growth

There is a critical line at 2900 m a.s.l.; i.e., there are markedly different controlling factors for the seasonal patterns of stem diameter growth below or above this elevation. The growth initiation was controlled by air temperature at both low (2500 m and 2700 m a.s.l.) and middle (2900 m a.s.l.) elevations, whereas it was controlled by both air temperature and soil temperature at high elevations (3100 m and 3300 m a.s.l.). The thresholds of air temperature at low elevations ranged from 8.4 °C to 10.7 °C, and the thresholds at both middle and high elevations ranged from 3.5 °C to 6.8 °C for air temperature and from −0.3 °C to −0.1 °C for soil temperature (Table 4).

**Table 4.** Thresholds of environmental factors (mean air temperature ($T_a$), mean soil temperature ($T_s$) and volumetric soil moisture content ($M_s$) of the 0 to 60 cm soil layer) for seasonal patterns of stem growth of Qinghai spruce trees in the Pailugou watershed.

| Growth Characteristics | Environmental Factors | Elevation (m a.s.l.) | | | | |
|---|---|---|---|---|---|---|
| | | 2500 | 2700 | 2900 | 3100 | 3300 |
| Initiation | $T_a$ | 10.62 ± 4.41 a | 9.38 ± 3.93 a | 8.40 ± 3.86 a | 6.79 ± 3.79 ab | 3.50 ± 2.80 b |
| | $T_s$ | 1.66 ± 0.87 b | 2.70 ± 0.37 a | 0.06 ± 0.37 c | −0.18 ± 0.32 c | −0.25 ± 0.28 c |
| | $M_s$ | 16.19 ± 0.41 e | 18.55 ± 0.09 d | 50.58 ± 0.02 b | 51.85 ± 0.04 a | 43.18 ± 0.07 c |
| Cessation | $T_a$ | 7.61 ± 1.80 b | 11.90 ± 2.65 a | 5.19 ± 1.46 c | 12.46 ± 0.85 a | 2.76 ± 1.52 d |
| | $T_s$ | 5.33 ± 0.83 c | 7.85 ± 0.31 a | 2.46 ± 0.59 d | 5.98 ± 0.50 b | 1.68 ± 0.57 e |
| | $M_s$ | 21.17 ± 0.58 d | 21.44 ± 1.91 d | 51.52 ± 0.33 b | 52.49 ± 0.54 a | 43.67 ± 0.50 c |

Note: The different lowercase letters indicated significant differences at different elevations for each environmental factor ($p < 0.05$).

The growth cessation at low elevations coincided with the occurrence of high soil temperature (5.33–7.85 °C) and low soil water (21.1–21.5%), whereas water stress was absent at the end of stem growth at both middle and high elevations (Table 4).

### 3.4. Impacts of Environmental Factors on the Daily Stem Growth Rate

In stages 1 and 3 (i.e., the beginning and end stages), the $G_r$ at all elevations was significantly positively correlated with $T_a$ and $T_s$, with correlation coefficients greater than 0.4 (Figure 4). There was, however, contrasting data concerning soil moisture in stage 1; the $G_r$ for each elevation was significantly positively correlated with $M_s$ in 2015, with correlation coefficients greater than 0.9, while it was significantly negatively correlated with $M_s$ in 2016. In stage 3 of 2015, $G_r$ was significantly positively correlated with $P$ at 2700 m and 3100 m a.s.l., but it was significantly negatively correlated with $P$ at other elevations.

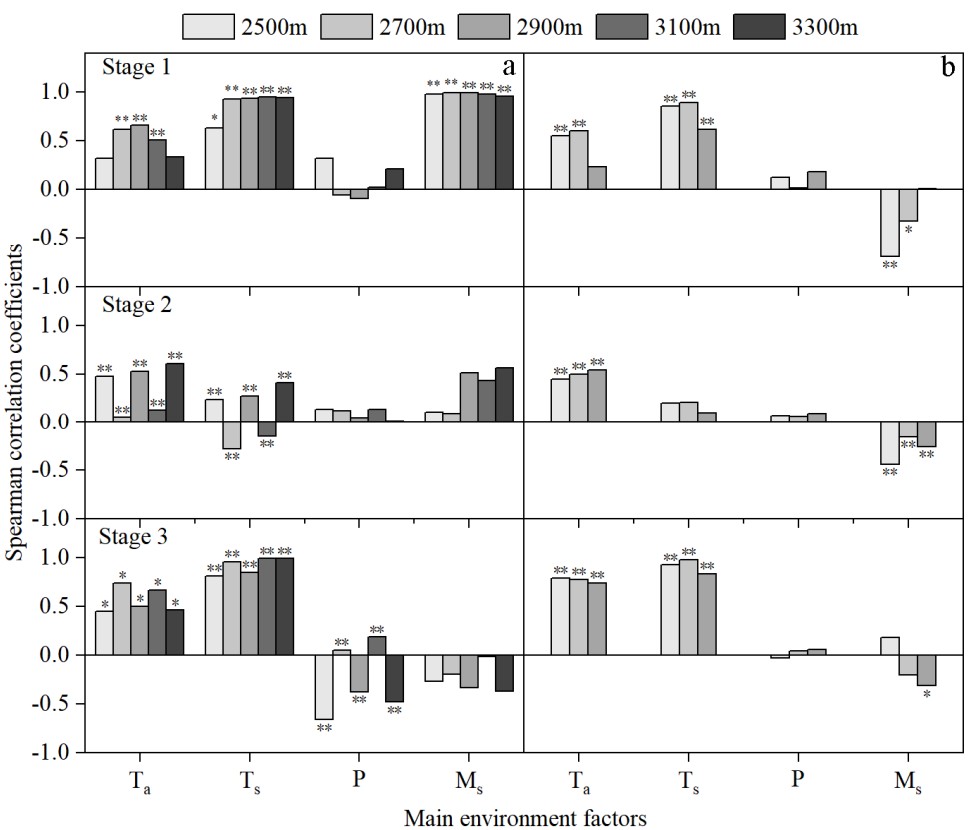

**Figure 4.** Spearman correlation coefficients of the daily rate of stem growth ($G_r$) and air temperature ($T_a$), precipitation ($P$), volumetric soil moisture content ($M_s$) and soil temperature ($T_s$) at the 0 to 60 cm soil layers in stages 1, 2, and 3 in 2015 (**a**) and 2016 (**b**). Values marked ** and * indicate that they are significant at the 0.01 and 0.05 levels, respectively.

In stage 2 (the rapid growth stage), the $G_r$ at all elevations was positively correlated with $T_a$ and $P$ in 2015 and 2016. There was, however, contrasting data concerning $M_s$; the $G_r$ was positively correlated with $M_s$ in 2015, but it was negatively correlated with $M_s$ in 2016. Similarly, the $G_r$ was significantly negatively correlated with $T_s$ at 2700 m and 3100 m a.s.l. in 2015, but positively correlated with those in 2016 (Figure 4).

## 4. Discussion

### 4.1. Impacts of Environmental Factors on Seasonal Growth Patterns across Different Elevations

It was found that the growth initiation occurred later with increasing elevation in the study area. Since temperatures are the most important variables along the elevation gradient [28–30], this is due to the lower temperatures at higher elevations. This was consistent with our hypothesis that growth initiation was controlled by temperature at all elevations. Since temperatures can affect cambial activity by increasing the rate of cell division and the number of xylem produced [14,33]. Related studies reported that the growth of Qinghai spruce stems was generally initiated when the mean $T_a$ was maintained

above 5 °C or when the mean $T_s$ begin to increase steadily past 0 °C [23,24]. In this study, the growth initiation thresholds of mean $T_a$ and $T_s$ of the 0 to 60 cm soil layer was above 3.5 °C and 0 °C, respectively.

Our study showed a significant difference in growth cessation across elevations, with 35 days earlier at 2700 m a.s.l. compared with the middle and high elevations in 2015. This indicated that the controlling factors of growth cessation were different across elevations. For middle and high elevations, temperatures and soil moisture were very different, but the timing of growth cessation was similar. This implied that the growth cessation was probably governed by photoperiod and cell differentiation [15,18,34]. For low elevations, the growth cessation was mainly affected by soil moisture [20,35,36]. The coexistence of high temperatures and low $M_s$ seemed to trigger the growth cessation at 2700 m a.s.l., leading to a continuous soil water deficit since mid-July. Conversely, the low $M_s$ did not restrict the growth cessation at 2500 m a.s.l., mainly due to the unique microclimate conditions at the bottom of the ditch and approximately 5 m from the river. Thus, the insufficient rainfall and low $M_s$ in August 2015 compared with 2016 induced an earlier cessation of stem growth at low elevations, but not at middle and high elevations.

Several studies reported that the maximum growth rate for conifers in cold environments is synchronized with maximum day length [18,37]; however, some studies reported that it may synchronized not only with maximum day length but also with optimal soil water and maximum solar radiation [20,21,24]. In our study, the maximum growth rate occurred in July at all elevations, which was about 20 days later than the day of maximum daylength. At the summer solstice in both years, the study area experienced insufficient rainfall, low $M_s$ and extended periods of sunny conditions. Although maximum day length coincided with high temperatures and solar radiation, stem growth rate was not at its maximum due to lower $M_s$. Therefore, the maximum growth rate may synchronize not only maximum day length and suitable weather condition, but also with optimal soil water.

Elevation gradients allow for natural space-for-time/warming experiments to know the mechanism of seasonal growth in response to future climate changes [14,15]. Several studies reported that tree growth at high elevations was sensitive to temperature, and an increasing temperature could increase tree growth at high elevations [38,39]. However, an increasing temperature reduced tree growth at low elevations, mainly due to suffering from temperature-induced drought stress [7,40,41]. In this study, annual stem growth and growing season duration at low elevations were determined by a combination of spring temperature and autumn rainfall, while high elevations were determined by spring temperature. In the context of future warming, a warmer autumn has no effect on stem growth at higher elevations, but lower elevations may lead to drought stress and early cessation of growth. In contrast, spring warming is positive for stem growth in the whole forest zone. When soil temperatures reach above 0 °C, the snow and frozen soil melts, soil moisture content improves [24]. Consequently, soil water is sufficient when growth begins, and temperature is the main factor affecting stem growth. Spring temperature increases, stem growth for the whole forest zone begins earlier, and growing season duration is extended, which is good for tree growth and carbon sequestration of forest ecosystems in semi-arid mountains.

### 4.2. Impacts of Environmental Factors on Daily Stem Growth across Different Elevations

During the growing season, the daily stem growth was affected by both weather conditions and soil factors [20,23,24], and intra-annual stem growth of several coniferous species had a positive effect on temperature [42–44]. In this study, the $G_r$ at all elevations was also positively correlated with $T_a$ and $T_s$ at different growth stages in both 2015 and 2016, except for 2700 m and 3100 m during stage 2 of 2015 (Figure 4). During stage 2, the $T_a$ was high and peaked (Figure 2), and the increase in $T_a$ may have increased soil evaporation and tree transpiration [45]. Moreover, $M_s$ decreased and reached its minimum value; thus, the high temperature and low $M_s$ affected tree growth. In addition, Spearman correlation coefficients between $G_r$ and $T_a$ at 2700 m and 3100 m a.s.l. were lower than they were at

2500 m, 3100 m and 3300 m a.s.l. in stage 2 (Figure 4). These results further confirmed that high temperature caused water stress and led to stem shrinkage [24].

In this study, the $G_r$ at all elevations was positively correlated with $M_s$ in stages 1 and 2 in 2015. These results are in agreement with those of Jiang et al. [21], who found that the stem growth of *Platycladus orientalis* (L.) Franco was significantly positively related to $M_s$. However, the $G_r$ was negatively correlated with $M_s$ in stages 1 and 2 in 2016. This interannual difference may be caused by differences in soil conditions. When the soil water content is high due to precipitation, the soil temperature decreases, and root respiration becomes inhibited, which may affect stem growth [23]. Similarly, the $G_r$ at all elevations was negatively correlated with $M_s$ in stage 3 in both 2015 and 2016. This was partly because the soil in the Qinghai spruce plots begins to freeze in late October every year [46]. When the soil is frozen, tree roots are unable to take up water, which may explain the very low $G_r$, which was less than 2 $\mu m \cdot day^{-1}$ in stage 3.

The correlations between the $G_r$ and main environmental factors varied with growth stages (Figure 4). However, quantitative the relationships describing the effects of environmental conditions on stem growth were not available in this study due to limited data. Under changes of environmental conditions in the future, especially the high temperature and drought events, these changes will affect intra-annual stem growth. Therefore, further research should be encouraged to quantify the effects of weather and soil factors on stem growth, and to study the effect of seasonal drought on intra-annual stem growth. These results will provide a basis for further understanding of the controlling mechanisms of growth process.

*4.3. Implications for Forest Management*

Temperature and soil water availability is the important influencing factors for tree growth in mountain forests [21,32], but high temperatures and drought events will continue to increase in the future [1], which may further affect mountain forests [47,48]. Qinghai spruce is an important tree species for soil and water conservation in the Qilian Mountains. Additionally, thus, it is the important to maintain tree growth of this species. In fact, trees at low elevations were subjected to soil moisture deficit in case of insufficient rainfall, and they had a rather narrow growth. Previous studies also found that Qinghai spruce at low elevations suffer from water deficit, and tree growth decline [7,20]. To mitigate drought stress at low elevation, tree growth can be maintained by appropriately reducing stand density in future forest management. In addition, trees at middle and high elevations may grow faster due to warmer spring temperatures, and the optimal elevations for Qinghai spruce growth is expected to shift upward in the next few decades [49], so the middle and high elevations are more beneficial to the migration and survival of Qinghai spruce.

## 5. Conclusions

This study on the seasonal patterns of Qinghai spruce growth at different elevations in the Qilian mountains of northwestern China showed that growth initiation occurred later as the elevation increased in both years, which was controlled by temperature; however, growth cessation presented large elevational differences, and cessation occurred much earlier at lower elevations than at higher ones, especially at 2700 m a.s.l. in 2015, which was mainly due to seasonal drought since mid-July. Additionally, there was a "unimodal" trend of annual cumulative growth as the elevation increased, with a peak at 2900 m a.s.l. (1520 μm in 2015 and 1670 μm in 2016), which was 2.1–4.4 times that at other elevations. This difference in stem growth resulted from the varying temperature and soil moisture in the growing season, namely, an insufficient water supply in autumn at low elevations and low spring temperatures at high elevations that collectively may result in a shortened growing season and a rather narrow annual growth at low and high elevations. These results provide a phenological basis for predicting the future adaptation of Qinghai spruce forest in semi-arid area under climate changes.

**Author Contributions:** Conceptualization, Y.W. (Yanfang Wan), P.Y. and Y.W. (Yanhui Wang); instrument installation, L.Z., X.L. (Xiande Liu) and S.W.; investigation, Y.W. (Yanfang Wan), X.L. (Xiaoqing Li), L.Z., X.L. (Xiande Liu) and S.W.; formal analysis, Y.W. (Yanfang Wan) and X.L. (Xiaoqing Li); writing—original draft preparation, Y.W. (Yanfang Wan); writing—review and editing, Y.W. (Yanfang Wan), P.Y., Y.W. (Yanhui Wang), B.W. and Y.Y. All authors have read and agreed to the published version of the manuscript.

**Funding:** This work was financially supported by the National Natural Science Foundation of China (U21A2005, U20A2085, 41971038, 32171559) and the Central Public-Interest Scientific Institution Basal Research Fund of Chinese Academy of Forestry (CAFYBB2021ZW002, CAFYBB2020QB004).

**Data Availability Statement:** Not available.

**Acknowledgments:** We thank Ming Jin, Wenmao Jing, Weijun Zhao, Jian Ma, and other staff of the Academy of Water Resource Conservation Forests of Qilian Mountains in Gansu Province for their assistance in the field work.

**Conflicts of Interest:** The authors declare no conflict of interest.

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
