# Peer review of "Divergent Seasonal Patterns of Qinghai Spruce Growth with Elevation in Northwestern China"

_forests, doi:10.3390/f13030388_

Round 1

Reviewer 1 Report

The authors present a study about  stem diameter growth across different elevations. From my understanding, the technical parts of this work were done carefully and correctly. Moreover, I appreciate the effort the authors collecting the data. The research question is a very interesting and original topic. However, this article has some weaknesses that I list below:

  • Title is not appropriate: long and not clear: must to be simplified.
  • Introduction is very short and more references and details must be added to improve the topic description and the framework. To describe in detail the novel aspects of the study
  • Hypothesis are lacking could be improved.
  • Structure of methods is strange. Please, provide: study area description, experimental design, sampling protocol.
  • Some important international references are lacking.
  • Future research and recommendations are not indicated.
  • There are important elements that must be more strongly highlighted in this paper:

-          What are the original elements of this research?

-          Need to study this subject?

-         Implications for forest management?

At this stage, I propose the authors to consider these suggestions in a moderare revision, and request the editor not to accept the manuscript until and unless the authors make the changes. Congratulations to the authors, they have realized a good job in this original research. I have also provided the following specific comments that would help to improve the quality of this manuscript.

Specific comments

L49. Develop more the climate change context and the introduction, as well as the factors influencing gorwthé

L49. About climate change add this reference:

Hof, A. R., Montoro Girona, M., Fortin, M. J., & Tremblay, J. A. (2021). Using Landscape Simulation Models to Help Balance Conflicting Goals in Changing Forests. Frontiers in Ecology and Evolution, 818.

L51. Add this reference:

Ameray, A., Bergeron, Y., Valeria, O., Montoro Girona, M., & Cavard, X. (2021). Forest Carbon Management: a Review of Silvicultural Practices and Management Strategies Across Boreal, Temperate and Tropical Forests. Current Forestry Reports7(4), 245-266.

L57-67: This information is better in the study area. In the introcution you must to present the state of knowledge and the gaps of knowledge.

L74. Add hypotheses

L101. Replace this title for “experimental design”

Figure 1. Coordenates are lacking. Not good quality. Add pictures from the sites to show the changes in the gradient.

Figure 2,3,4,5: great!

L412: Add it:

Achim, A., Moreau, G., Coops, N. C., Axelson, J. N., Barrette, J., Bédard, S., ... & Montoro Girona, M. (2021). The changing culture of silviculture. Forestry: An International Journal of Forest Research, cpab047.

Pappas, C., Bélanger, N., Bergeron, Y., Blarquez, O., Chen, H. Y., Comeau, P. G., ... & Kneeshaw, D. (2022). Smartforests Canada: A Network of Monitoring Plots for Forest Management Under Environmental Change. In Climate-Smart Forestry in Mountain Regions (pp. 521-543). Springer, Cham.

Discussion

  • It is a very good discussion. However:
    • Implications for forest management are lacking
  • Propose future research

Author Response

Dear Sir/Madam,

    Thank you so much for your kind encouragement on our manuscript entitled “Divergent Seasonal Patterns of Stem Diameter Growth of Qinghai Spruce across Different Elevations in the Qilian Mountains, Northwestern China” (Forests-1544622) and sending us the review on our manuscript.

    We greatly appreciate you for your valuable comments and constructive suggestions and have carefully addressed these points in the new version.

    A point-to-point response to the comments has been detailed in our Response Letter, in which the comments are written in italics followed by our responses in regular text. In addition, the changes or revision were marked with red text in the manuscript.

    Now, we are resubmitting the latest vision of our manuscript to you.

Best wishes,

Pengtao Yu

Reviewer 2 Report

The manuscript is interesting and brings new findings related to stem diameter increment with regard to environmental factors. Although the paper is well written, still some minor changes would be changed.   

Specific comments:

Abstract

 the section is too long, hence, I recommend shortening the part showing own findings (by circa one third). The authors would not explain all results here, just those most relevant.

Introduction

the section is too much focused on the specific region where the study was performed. I wish the authors to explain also broader (international) aspects of the study. In fact, the main part of the second paragraph would be rather included in the Material and Methods section. Moreover, the authors would be careful not show the same information about the region (first in Introduction, second in Material and Method).

Material and Methods      

The authors inform about the elevations of the plots tree times (Figure 1, Tables 1 and 2). It should be shown just only once (perhaps in Table 1).

Table 2. Since number of sampling trees was the same on the all plots, it is rather uncommon showing the same number five times. I suggest erasing the columns dedicated to number of measured trees. The information might be mentioned in the main text or as a part of the Figure caption.     

Figure 2. I suggest to modify the upper diagrams for colour version. Moreover, I do not like the Legend which shows the form of expression for precipitation.

Figure 4. I think the fitting (excepting the first plate) is rather speculative and relationship is just vague. If the authors want to keep all diagrams in the Figure 4, their interpretations would be very modest and with high level of uncertainty (the same in the Discussion section).

Table 4. I believe that the first column would be split into two separate columns. The first might show two “Growth characteristics” and the second “Environmental factors”.

Discussion     

The section is rather nice to me. On the other hand, I feel like contains too much repetitive information from the Results section (especially in 4.1). It would be little bit reduced. Moreover, I am missing broader (international) aspects of the outputs, please try to add some more parallels with foreign research works and potential implementation in larger extent.

Conclusions

The same complain as for the Discussion section. The text is too much focused on the local issues. Here, the broader context plus better generalization of findings would be added.    

Author Response

Dear Sir/Madam,

    Thank you so much for your kind encouragement on our manuscript entitled “Divergent Seasonal Patterns of Stem Diameter Growth of Qinghai Spruce across Different Elevations in the Qilian Mountains, Northwestern China” (Forests-1544622) and sending us the review on our manuscript.

    We greatly appreciate the reviewers for your valuable comments and constructive suggestions. We have carefully addressed these points in our revised manuscript and a point-to-point response to the comments has been detailed in our Response Letter, in which the comments are written in italics followed by our responses in regular text. In addition, the changes or revision were marked with red text in new version of the manuscript.

    Now, we are resubmitting the latest vision of our manuscript to you.

Best wishes,

Pengtao Yu
